# *Data Passports*: Confidentially Provable Provenance for Onboarding Verifiable ML

## Abstract

Recent advances in ML have leveraged Zero Knowledge Proof protocols to enable institutions to cryptographically commit to a dataset and subsequently prove, to external auditors, the integrity of training and the trustworthiness of the resulting model on the committed data, all while protecting model confidentiality. Such approaches guarantee that the training algorithm which produced a model was computed correctly, but remain vulnerable to pre-commitment data tampering. This is because even if the training algorithm is executed faithfully, an institution can bypass the audit by manipulating the training data. Likewise, data generators may degrade a model's utility via data poisoning.

To address this, we introduce **tamper-proof *Data Passports*** that bind data to verifiable and confidential proofs of authenticity. We leverage Trusted Execution Environments to issue a certificate of authenticity or 'passport' for each data point produced by a generating process. The generating process passes the data and passport to the institution. Then, the institution uses a zero-knowledge proof to verify the validity of the passports to an auditor, as an onboarding step for downstream proofs of training integrity and model trustworthiness. This unlocks cryptographic verification of data provenance throughout the ML pipeline.

Our experiments demonstrate that we can create tamper-proof passports for images taken by users on their smartphones with a very negligible overhead. Agnostic to data size, a passport can be created at capture time in only 230 ms and consumes just 4.8 KB; thus, it has minimal impact on compute, storage and network usage.

## 1 Introduction

Institutions often collect data from users to train and fine-tune models, subsequently offering machine-learning-as-a-service in critical applications such as healthcare, criminal justice, hiring, and finance. Given risks (Angwin et al., 2016; Seyyed-Kalantari et al., 2021; Buolamwini & Gebru, 2018) associated with both training and inference, governments have begun regulating AI to ensure that these technologies are developed and deployed responsibly and ethically (Biden, 2023). Public verification called for so that an independent external party (*auditor*) from outside of the institution verifies whether a model is trustworthy and upholds privacy (Dwork, 2006), fairness (Dwork et al., 2012; Hardt et al., 2016), and other objectives mandated by law or societal values. In such settings, it is essential to protect the confidentiality of users' data and the intellectual property of the institutions. For example, institutions are not allowed to share users' medical data, as it is protected by legislation such as HIPAA in the United States and PIPEDA in Canada.

To address these needs, the institution provides various form of confidential proofs to an auditor–namely confidential proof of training (Abbaszadeh et al., 2024) to ensure the integrity of the training stage and prove the institution's ownership of the model, confidential proof of fair training (Shamsabadi et al., 2023; Franzese et al., 2024; Yadav et al., 2024; Zhang et al., 2025) to prove that the model satisfies fairness constraint, and confidential proof of differentially-private training (Shamsabadi et al., 2024) to establish public trust in the training process and their model. Such confidential proofs are constructed by leveraging zero-knowledge proofs (Goldwasser et al., 2019; Goldreich et al., 1991), allowing the institution to commit to its training dataset and subsequently prove the correctness of the agreed-upon training algorithm on the committed data to the auditor while preserving the confidentiality of all the training data, intermediate model updates, and final trained model thanks to the hiding property of the commitment scheme.

Although the binding guarantee of the commitment scheme prevents the institution from manipulating the data being committed without being caught by the auditor, two critical *data vulnerabilities* remain: *(i)* institutional data manipulation–a malicious institution can still *manipulate user data before committing* to it; and *(ii)* user data manipulation–a malicious user can manipulate data before sharing it with the institution. These vulnerabilities reflect real-world incentives. Institutions may seek to shape model behavior for profit or strategic advantage in practice. For example, an insurance company might train a model on skewed data to falsely justify rejecting legitimate claims (Napolitano, 2023) while passing fair training verification. Similarly, users may deliberately poison data to introduce targeted mistakes in the model's behavior (Carlini et al., 2024).

Such data vulnerabilities cannot be addressed by *post-hoc* privacy-preserving input validation approaches (Bell et al., 2023; Duddu et al., 2024) as they can only verify whether the data satisfies constraints such as $l_p$ bounds. A malicious party can still manipulate data while remaining in for example $l_p$ constraint to be unnoticed. Therefore, it is extremely hard to accurately detect manipulated data post-hoc based on content analysis. To address these issues, we propose to proactively generate a cryptographic passport of source authenticity directly on the user's device at the time of the data generation. We introduce *tamper-proof Data Passports*, a framework for confidentially verifiable data provenance that enhances data generation with a certificate of provenance information. We utilize Trusted Execution Environments (TEEs) (Liu et al., 2024; 2022; tru, a) to construct and sign provenance information for data generated by the user. Tamper-proof *Data Passports* contain the source origin of the data and any post-processing histories, enabling the auditor to verify the authenticity and integrity of data at the time of its generation without violating data confidentiality and without allowing any party to silently manipulate the data.

Our proposed passport-based commitments can be combined with existing confidential proof of training to fix their data vulnerabilities by preventing malicious institutions and users from manipulating training data. The institution commits to an authenticated dataset and proves in zero knowledge the correctness of all *Data Passports* before allowing the data to be included in training, ensuring that only authenticated data without any modification is used. In addition, our tamper-proof passports offer a second key advantage: establishing a trusted reference dataset for facilitating efficient ZKP certification of training through local computations. Proving the entire training process with heavy cryptographic machinery would be prohibitively expensive in most practical settings. *Data Passports* allow the institution to carry out the computationally-intensive training *locally*, while only the fairness or calibration (Rabanser et al., 2025) evaluation needs to be performed in ZK on this authenticated reference dataset and verify that the final locally obtained model meets fairness or calibration criteria. Finally, *Data Passports* are beneficial to institutions against malicious users for both inference and training. *Data Passports* can be combined with ZKPs of Correct Inference (Weng et al., 2021) to prove that inference was computed correctly given an ML model and authenticated data, thus preventing the model's susceptibility to adversarial examples created by users. *Data Passports* allow institutions to prevent data-poisoning risks introduced by users (Carlini et al., 2024), such as injecting backdoors Qi et al. (2023); Gu et al. (2019); Zhu et al. (2025), and remove manipulated data from users who seek to alter the performance of the model during training.

In this paper, through a co-design of TEE and ZKP, we introduce an authenticated user-generated dataset equipped with confidentially verifiable passports. These *Data Passports* enable diverse trustworthiness audits and provide strong assurance to the public that no parties–including users, institutions, and auditors–can silently manipulate the data, while protecting its confidentiality. We implement *Data Passports* and evaluate User-side TEE efficiency and Institution-side ZKP efficiency. On Samsung Galaxy S20 Plus and Google Pixel 6 Pro smartphones, we measure CPU, memory, and battery consumptions of user-generated data as well as the storage requirements and execution time of passport creation. We highlight the following contributions:

- We introduce a new threat model of *self-poisoning attacks* where the institution manipulates the training data *itself* to pass an audit, highlighting data vulnerabilities in existing ZKP verifications.

- We propose a framework for user-generated data with a *confidential passport created at the time of generation*–analogous to a passport issued to a newborn at birth. Confidential passports encode provenance information– how the data was generated and what modifications were made to it.

- We show that passport-based data commitments are mutually beneficial for all parties, complement confidential auditing, and mitigate data vulnerabilities.

- We implement TEE-based *Data Passports* on user devices and ZKP verification on the institution side. *Data Passports* introduces very negligible overhead on user devices. We submitted a demo of *Data Passports* as part of the supplementary material.

## 2 BACKGROUND

In this section, we introduce background concepts in Trusted Execution Environments (TEEs) and Zero Knowledge Proofs (ZKPs).

**Background on TEE.** A TEE, sometimes referred to as an enclave, is a secure area of a processor that provides an isolated computing environment, separate from traditional rich runtime environments such as an Operating System (OS). A TEE enforces strong hardware-backed guarantees of confidentiality and integrity for the code and data it hosts. These guarantees are achieved through enforcements such as dedicated memory accessible only to the TEE. Hardware isolation ensures that even a fully compromised OS cannot access (either read or write) orx tamper any code or data residing inside the TEE. In addition to this, TEEs expose unique hardware primitives such as secure boot and remote attestation to ensure only trusted code is loaded into TEE and external parties being able to verify the integrity of TEE. TEEs are widely available across various system and processor architectures, such as TrustZone arm (b) and Realms arm (a) for ARM, Intel SGX int (a), TDX int (b), and AMD SEV amd for x86, and WorldGuard ris, KeyStone Lee et al. (2020), and Penglai Feng et al. (2021) for RISC-V. In this work, we adopt ARM TrustZone for our prototype, as it is widely deployed in commodity devices such as ARM-based smartphones tru (b). As an alternative, we also consider Android StrongBox and, a Hardware Security Module (HSM) that has gained popularity in recent years on select Android devices. Popular StrongBox solutions include Samsung Knox Vault sam (a) and Google Titan M/M2 pix. The main difference between such a HSM and TEE is that HSM is built specifically for cryptographic operations, where TEE is designed to execute code securely. Since our passport generation process only involves cryptographic operations, both TEE and HSM can be utilized here. For user devices in our evaluations, we use two representative devices–Samsung Galaxy S20 Plus and Google Pixel 6 Pro– both of which support TrustZone and StrongBox.

**Background on ZKP.** In this work, we use ZKPs to verify the validity of data passports to the auditor, without revealing information about the data. A ZKP is a cryptographic protocol for verifying properties of hidden data. A ZKP takes place between two parties, a prover $P$ and auditor $V$. $P$ has a string of hidden data called a 'witness' $w$, and they would like to prove to $V$ that $w$ satisfies some property. The property is encoded as a circuit $C$, which is known to both parties. $P$ and $V$ can execute a ZKP protocol to prove whether $C(w) = 1$. A secure ZKP protocol has the following properties: (i) *Completeness* – for all $w$ such that $C(w) = 1$, $P$ can use the ZKP protocol to prove that $C(w) = 1$. (ii) *Soundness* – if $C(w) \neq 1$, $P$ cannot use the protocol to falsely convince $V$ that $C(w) = 1$ even if $P$ performs arbitrary malicious behavior during protocol execution. (iii) *Zero-Knowledge* – $V$ learns no information about $w$ other than what is implicitly revealed by knowing that $C(w) = 1$, even if $V$ performs arbitrary malicious behavior during protocol execution.

## 3 *Data Passports*

Our framework confidentially verifies that an institution owns data from a set of certified data sources (users) without any data manipulation while preserving the confidentiality of the data.

### 3.1 PARTIES

As visualized in Figure 1, we achieve this through a co-design of ZKP and TEE between four parties: Certificate Authority $\mathcal{CA}$, Certified Data Sources $\mathcal{S}$, Institution $\mathcal{I}$, and Auditor $\mathcal{V}$:

- **Certificate Authority $\mathcal{CA}$.** A trusted third party that certifies sources of data. Similar to a certificate authorities that register trusted web domains, $\mathcal{CA}$ obtains confirmation that the data is generated by an authenticated source. $\mathcal{CA}$ then publishes a list of privacy-preserving certificates, allowing other parties to verify whether data came from a trusted source. In this work, we realize $\mathcal{CA}$ as the *issuer of tamper-proof TEE devices*, but it could also be a government body that verifies identity

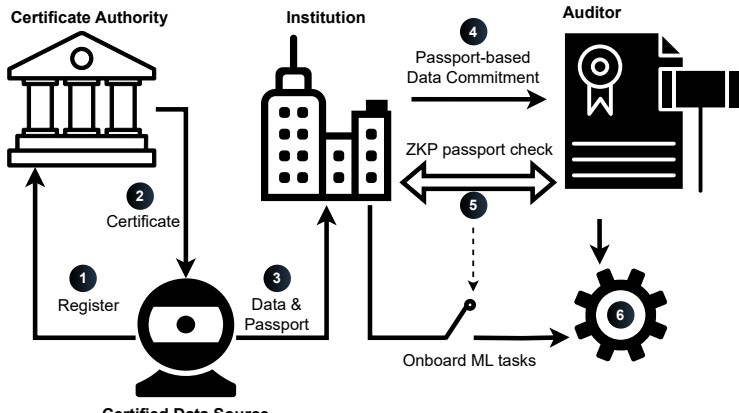

Figure 1: **Block diagram of tamper-proof *Data Passports* for ML onboarding.** ①-②: A data source registers with a certificate authority to obtain certification. ③: The certified data source generates data and shares it with the institution, along with its passport. ④-⑤: The institution performs passport-based data commitment to enable zero-knowledge verification by the auditor. Once the auditor validates the passports, the institution is authorized to onboard ML tasks ⑥.

and demographic status of individuals, or a professional organization that certifies reputable data vendors.

- **Certified Data Sources** $\mathcal{S}$ generates the data. In this work, we realize $S$ as a set of tamper-proof TEE devices, but they could also be instantiated as certified data brokers or individuals.

- **Institution** $\mathcal{I}$ who wishes to confidentially verify model properties with certified data.

- **Auditor** $\mathcal{V}$ who wishes to verify data provenance and then data-dependent properties of $\mathcal{I}$'s model.

### 3.2 OVERVIEW

Our framework consists of six steps: steps (1-3) make sure that $\mathcal{S}$ provides the unmodified outputs of sensors, while steps (4-6) make sure that $\mathcal{I}$ performs a data-dependent operation (e.g., training, fine-tuning, auditing) with data from certified sources.

1. **Data source registration**. Each data source $S_i \in \mathcal{S}$ registers with the certified authority $\mathcal{CA}$, who confirms that they are a valid data source.

2. **Data source certification**. Each $S_i$ obtains a certificate authenticating data it generates.

3. **Data generation and passport creation**. Each data source $S_i$ generates a data point $\mathsf{data}_i$. Each $S_i$ is in possession of a $\mathsf{passport}_i \leftarrow \{\mathsf{Signature}_i, \mathsf{Certificate}_i\}$ containing provenance information. The certified $\mathcal{S}_i$ uploads $(\mathsf{data}_i, \mathsf{passport}_i)$ to $\mathcal{I}$.

4. **Passport-based data commitments.** $\mathcal{I}$ aggregates data $\mathsf{DataBase} = ||_{i=1}^{N} \mathsf{data}_i$ and passports $\mathsf{PassportBase} = ||_{i=1}^{N} \mathsf{passport}_i$ from $S$. $\mathcal{I}$ commits to $\mathsf{DataBase}$ and $\mathsf{PassportBase}$, and sends the commitments to $\mathcal{V}$.

5. **Zero knowledge data passport verifications**. $\mathcal{I}$ uses a zero-knowledge proof to verify to $\mathcal{V}$ that each passport contains provenance information of its corresponding data point.

6. **Onboarding ML**. $\mathcal{V}$ authorizes $\mathcal{I}$ to onboard ML tasks and use this authentic data to reliably and verifiably perform data-dependence operations–such as training a model or measuring the fairness of a model.

Next, we describe data generation and passport creation in TEEs and passport verifications in ZKPs, in details.

### 3.3 DATA GENERATION AND PASSPORT CREATION

We aim to construct a *confidentially verifiable* tamper-proof passport, $\mathsf{passport}$, for authentic data, $\mathsf{data}$, without revealing any information contained within the $\mathsf{data}$ itself.

---

**Algorithm 1:** *Data Source Registration* at manufacture time

---

**Input:** Certified Authority $\mathcal{CA}$, Data source $\mathcal{S}$
**Output:** Registered Data source $\mathcal{S}$

1: $\mathcal{CA}$ provisions device attestation keypair $(\text{pk}_a, \text{sk}_a)$ into $\mathcal{S}$
2: $\mathcal{CA}$ installs attestation certificate chain $\text{Chain}_a = [\text{Certificate}_a, ..., \text{Root}]$ on $\mathcal{S}$ at manufacture time
3: DeviceMetaData $=<$ "deviceId" : DeviceID, ... $>$
4: $\sigma_a = \texttt{Sign}_{\text{sk}_a}(\text{DeviceMetaData})$      ▷ $\mathcal{S}$ builds attestation statement
5: $\mathcal{S}$ sends $(\pi, \sigma_a, \text{Chain}_a)$ to $\mathcal{CA}$
6: **if** $\texttt{Verify}_{\text{pk}_a}(\text{DeviceMetaData}, \sigma_a) = 1$ and Freshness and Policy **then**
7:     $\mathcal{CA}$ registers $\mathcal{S}$ and opens a secure TLS connection for it

---

**Algorithm 2:** *Data Source Certification*

---

**Input:** Registered Data source $\mathcal{S}$
**Output:** Certificate, pk, sk

1: $(\text{pk}, \text{sk}) \leftarrow \textsf{KeyGen}()$      ▷ $\mathcal{S}$ generates a pair secret-public key
2: DeviceMetaData $=<$ "deviceId" : DeviceID, ... $>$
3: $\mathcal{S}$ sends pk and DeviceMetaData to $\mathcal{CA}$ through the secure TLS connection
4: certificate $\leftarrow \{\texttt{Sign}_{\text{sk}_{ca}}(\text{pk}\|\text{DeviceMetaData}), \text{pk}_{ca}\}$    ▷ $\mathcal{CA}$ signs it with its secret key
5: $\mathcal{CA}$ sends certificate to $\mathcal{S}$
6: **Return**: certificate and certified keypairs pk, sk

---

**Algorithm 3:** *Data Generation and Passport Creation*

---

**Input:** Certified kerpairs pk, sk, SHA256 Hash function
**Output:** $\{\textsf{data}, \textsf{passport}\}$

1: $\textsf{data}$      ▷ Generate data on $\mathcal{S}$
2: $\textsf{passport} \leftarrow \{\text{signature} = \texttt{Sign}_{\text{sk}}(\texttt{SHA256}(\textsf{data})), \text{certificate}\}$    ▷ Create Passport on $\mathcal{S}$

---

We leverage TEEs in the user's device to issue such passports by incorporating provenance information at the time of data generation. A TEE provides a secure, isolated execution environment for a security-critical program. We assume the availability of a TEE on user devices where data is generated. This TEE can be used to cryptographically sign the data generated by the device.

We define a passport as $\textsf{passport} = \{\text{signature}, \text{certificate}\}$, where signature is a digital signature computed over data using a secret key associated with a certificate provisioned to TEE:

1. Algorithm 1 registers data sources. $\mathcal{CA}$ attests each $\mathcal{S}_i$ device to ensure it is an authentic (untampered and genuine) device secured by TEE against malicious users. This is done through performing checks on the integrity of the OS and Application (e.g., Camera), as well as other libraries and applications residing on the device. Examples include Google Play Integrity API (goo) and Samsung Knox Attestation (sam, b).

2. Algorithm 2 certifies data sources. The certified $\mathcal{S}_i$ device securely (in TEE) generates a new pair of secret key sk and public key pk, $(\text{sk}_i, \text{pk}_i)$, for signing the data and verifying the signature, respectively. Note that $\text{sk}_i$ is hidden but $\text{pk}_i$ will be published through the certificate so that the auditor knows which sources are certified. The $\mathcal{CA}$ authenticates the new key pair sk by signing the public key and device metadata and issuing a certificate for this key, certificate.

3. Algorithm 3 captures data and creates a passport on certified sources. The certified $\mathcal{S}_i$ proceeds capturing data in TEE with a specific pre-defined public resolution. We support any sensing data, including photos, video, and audio. $\mathcal{S}_i$ uses the certified secret key sk to have a signature of the captured data in TEE using ECDSA with Secp256r1 curve, which is common and supported by devices in the real world (details in Section 4.2).

---

**Ideal Functionality $\mathcal{F}_{\text{ZKP-CI}}$**

**Participants:**

- Input Sources $S = ||_{i=1}^{N} S_i$
- Institution $\mathcal{I}$
- Auditor $\mathcal{V}$

**Public parameters:**

- circuit $C$
- list of certified public keys $\text{PK} = ||_{i=1}^{N} \text{pk}_i$

**Functionality:**

1. Each $S_i$ sends input $(\text{data}_i, \text{passport}_i)$
2. Send $\text{DataBase} = ||_{i=1}^{N} \text{data}_i$ and $\text{PassportBase} = ||_{i=1}^{N} \text{passport}_i$ to $\mathcal{I}$
3. $\mathcal{I}$ defines a subset of indices $Q \subseteq [1, N]$ and a set of inputs $\text{DataBase}' = \{\text{data}_i'\}_{i \in Q}$, sends $\text{DataBase}'$ and input $w$
4. If $\text{Vrfy}_{\text{pk}_i}(\text{data}_i', \text{passport}_i) \neq 1$ for any $i \in Q$, then send $\perp$ to $\mathcal{V}$
5. Otherwise, send `true` to $\mathcal{V}$ if $C(w, \text{DataBase}') = 1$ or send `false` to $\mathcal{V}$ if $C(w, \text{DataBase}') = 0$.

---

Figure 2: Ideal functionality for Zero-Knowledge Proofs with Certified Inputs (ZKP-CI).

### 3.4 PASSPORT-BASED DATA COMMITMENTS AND VERIFICATION

To capture our goals formally, we define an abstract cryptographic primitive called a *Zero-Knowledge Proof with Certified Inputs* (ZKP-CI). Zero-knowledge proofs are a well-studied cryptographic primitive, which ensures that a prover executes a publicly known circuit $C$ on a chosen input $w$, and sends its correct output to a verifier. A ZKP-CI modifies this primitive so that certified third parties can also provide inputs $X$ to the circuit, such that $X$ cannot be modified by the prover. The ideal functionality $\mathcal{F}_{\text{ZKP-CI}}$ is provided in Figure 2.

The functionality is realized by executing a standard zero-knowledge proof with a circuit verifying the signature associated with each data point, in conjunction with the arbitrary circuit $C$. This could be used to perform e.g. zero-knowledge proofs of correct training while preventing the input-based vulnerabilities of previous work. We estimate the performance cost of this addition in Section 4.2.

## 4 EXPERIMENTS

Our principle motivation for introducing tamper-proof *Data Passports* is to enable auditors to confidentially verify that institutions use authentic data generated by certified users without any manipulations. Next, we implement *Data Passports* and evaluate the novel aspects: i) authentic data generations with passports; and ii) zero-knowledge passport verifications. Therefore, we empirically validate efficiency in i) generating data and creating passports on the user device; and ii) performing zero-knowledge verification of *Data Passports* between institutions and auditors.

### 4.1 USERS CAN EFFICIENTLY GENERATE AUTHENTIC DATA WITH PASSPORTS

**Implementation**. We implement an Android-based prototype camera application. Using Hardware-backed KeyStore, the application is capable of using both ARM TrustZone and Android StrongBox to generate keys and provide a cryptographic signature for captured images. For an output image, the passport is embedded as the metadata of the JPEG file, as this can make our system seamlessly adopted by existing training pipelines. We use JPEG as it is a widely used standard format for images. We evaluated the Android-based prototype on two smartphones capable of both TrustZone

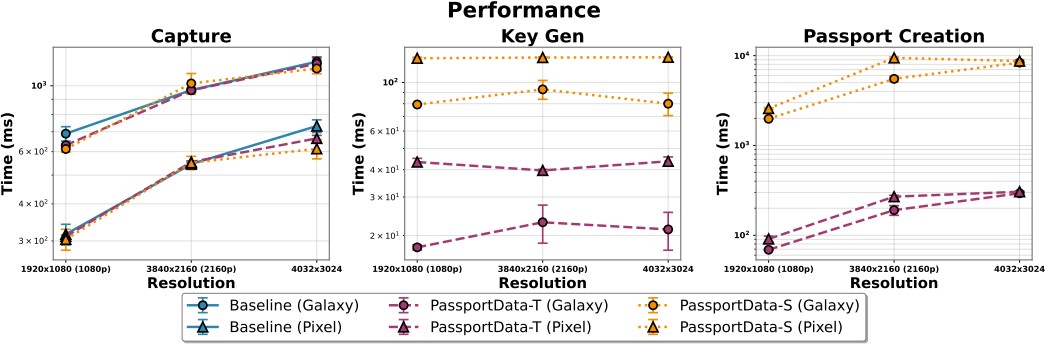

Figure 3: **A user can capture image of 1920x1080 (1080p) resolution with a passport in 715 ms.**

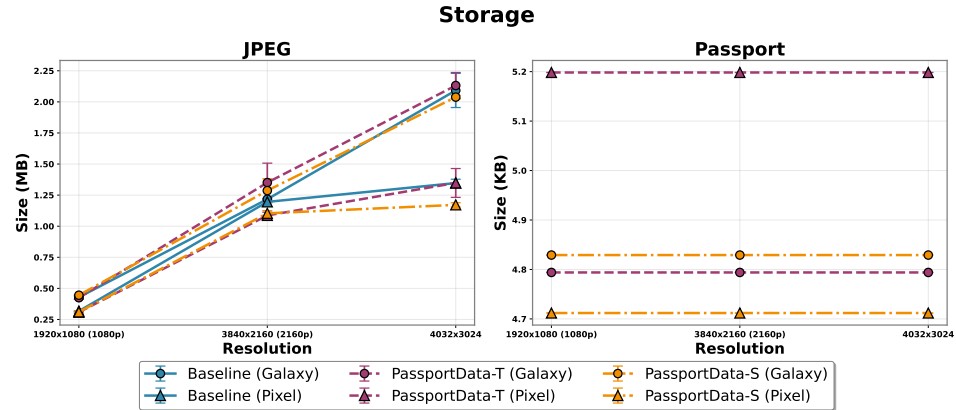

Figure 4: **An passport-enabled image with a resolution of 1920x1080 (1080p) has a size of 426 KB.**

and StrongBox: Samsung Galaxy S20 Plus and Google Pixel 6 Pro, where the former runs Android 13 and the latter runs Android 16.

We have two pipelines: PassportData-T and PassportData-S. They utilize different hardware primitives to execute the same passport generation sequence, where the former makes use of TrustZone backend, and the latter relies on StrongBox backend. In addition, we introduce a baseline pipeline, where it does not have any passport-related code and only takes a photo using the system camera API. We run all three pipelines on both smartphones across three different resolutions: 4032x3024, 3840x2160, and 1920x1080. We do want to notice that ML training-related workload usually involves a much lower resolution (e.g., 512x512); however, the lowest resolution that is supported by both of these two devices is 1920x1080. Furthermore, in order to learn how our system scales, we pick another two resolutions for comparison.

**Performance**. Figure 3 shows how each step in the three pipelines performs under different resolutions. In the left sub-figure, we can see that our system does not bring any overhead to the capture time itself. The middle sub-figure demonstrates that the key generation time remains the same on each hardware primitive across different resolutions. In the right sub-figure, the passport creation process scales linearly to resolutions on all hardware primitives. In terms of the potential effect brought by the performance overhead of our system, the PassportData-T pipeline introduces overhead (60 - 300 ms) that should not be noticeable by users, where the PassportData-S pipeline introduces noticeable overhead (2000 - 9000 ms), but is also good enough for offline use cases.

**Storage**. Figure 4 shows storage overhead. As illustrated by the left sub-figure, the JPEG data size does not get affected by our system. In the right sub-figure, we can say that the passport size remains unchanged on each hardware primitive across different resolutions. Overall, our system introduces negligible amount of storage overhead ($\sim$ 5 KB) compared to the size of JPEG images.

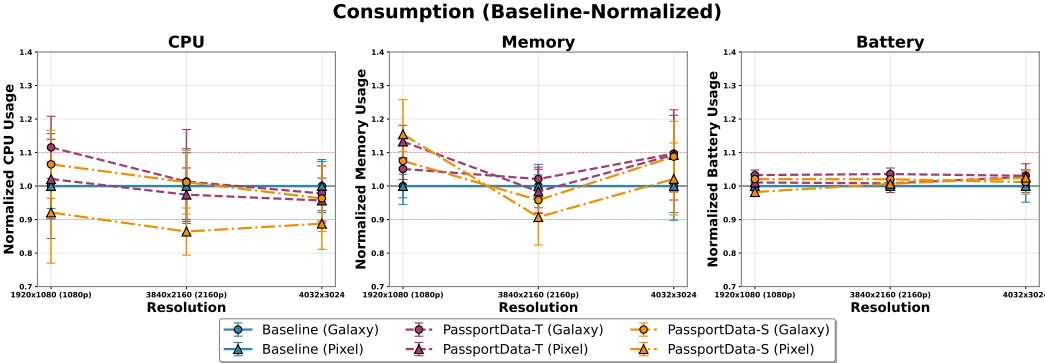

Figure 5: **While taking an image of 1920x1080 (1080p) resolution with a passport, the average consumption is about 18% CPU, 220 MB memory, and 4252 mW power.**

**Consumptions**. The three sub-figures in Figure 5, if we get rid of the errors (as we could only measure the entire system consumptions with many other running processes), we can clearly see that our system does not introduce any noticeable overhead in CPU, memory, or battery.

**Scalability**. In all steps of our system, only the hashing step during passport generation can be affected by changes in resolution, as lower resolution means a smaller file size and higher resolution means a larger file size. In both PassportData-T and PassportData-S on the two devices we evaluated with, time consumed in hashing is linearly related to image resolutions. Therefore, we can conclude that our system scales linearly in terms of performance overhead, and the storage overhead is constant, as we observed on various hardware primitives.

In summary, the PassportData-T pipeline has negligible performance overhead on both phones. The PassportData-S pipeline incurs moderate performance overhead, though this is acceptable as long as real-time usage is low. On the other hand, both pipelines have negligible storage overhead.

### 4.2 DATA PASSPORTS CAN BE CONFIDENTIALLY VERIFIED

After collecting all data and their passports, the institution first commits to the data (step ④, Figure 1) and then proves to the auditor that the passport is indeed consistent with the committed data (step ⑤, Figure 1). The institution performs the verification by generating a zero-knowledge proof of knowledge of ECDSA signature (steps 4 and 5 in Figure 2). This passport verification consists of two steps: 1) **hash verification:** institution proves that the hash digest of the image is consistent, and 2) **signature verification:** the signature of the hash is valid wrt to the signature verification key. Specifically, The institution generates a zero-knowledge proof that it knows the photo bytes $x_i$ whose hash digest $e_i$ both (i) equals $\text{hash}_{\text{sha256}}(x_i)$ and (ii) is consistent under the TEE's ECDSA public key $pk$ and signature $\sigma_i$ for all $i \in \{1, \dots, N\}$:

$$\exists\, x_i, e_i, pk_i\ :\ e_i = \text{hash}_{\text{sha256}}(x_i)\ \wedge\ \text{Vrfy}_{pk_i}(pk_i, e_i, \sigma_i) = 1.$$

We estimate the end-to-end cost of passport verification in zero-knowledge based on the ZK-ECDSA framework by Frigo & abhi shelat (2024). Frigo and shelat present a practical ZK framework that consists of highly optimized circuits of these two building blocks: (1) a *SHA-256 preimage* circuit over a binary field for the hash-consistency check, which corresponds to our **hash verification** step, (2) an *ECDSA-verify* circuit over P-256 that replaces modular inverses with a group-identity check for efficient verification, which corresponds to our **signature verification** step, and (3) a lightweight information-theoretic MAC to ensure the consistency of digest $e$ across the two fields. In our setting, we use their *SHA-256 preimage* circuit to prove $e = \text{hash}_{\text{sha256}}(m)$ and their *ECDSA-verify* circuit to prove $\text{Vrfy}_{pk}(e, \sigma) = 1$. This yields an end-to-end proof that the data passport is valid: the committed image $x$ is exactly the content whose digest was signed by the issuer's key and is consistent with its data passport.

We now estimate the cost of end-to-end passport verification for a single image. We extrapolate costs directly from the reported measurements in Frigo & abhi shelat (2024) since their implementation

Table 1: End-to-end cost of verifying a single data passport in ZK. Estimations are based on the prover time reported in Frigo & abhi shelat (2024). **#Blocks** is the total number of 64-byte blocks for each image. **Hash** is the prover time for hash verification, which costs $\approx 9$ ms per 64-byte block. **Total** is the end-to-end cost of passport verification, which adds a constant ECDSA-verify prover time ($\approx 60$ ms); the MAC-based cross-field consistency adds negligible overhead. All numbers reported are prover time, as verifier costs are negligible in comparison.

| Image Resolution | Device | TEE | File Size | #Blocks | Hash | Total |
|---|---|---|---|---|---|---|
| $4032 \times 3024$ | Galaxy S20+ | TrustZone | 2033.6 KB | 32,539 | 292.851s | 292.911s |
| | Galaxy S20+ | StrongBox | 2068.6 KB | 33,099 | 297.891s | 297.951s |
| | Pixel 6 Pro | TrustZone | 1214.6 KB | 19,434 | 174.906s | 174.966s |
| | Pixel 6 Pro | StrongBox | 1187.0 KB | 18,992 | 170.928s | 170.988s |
| $3840 \times 2160$ | Galaxy S20+ | TrustZone | 1189.2 KB | 19,028 | 171.252s | 171.312s |
| | Galaxy S20+ | StrongBox | 1206.7 KB | 19,308 | 173.772s | 173.832s |
| | Pixel 6 Pro | TrustZone | 1084.5 KB | 17,352 | 156.168s | 156.228s |
| | Pixel 6 Pro | StrongBox | 1097.9 KB | 17,566 | 158.094s | 158.154s |
| $1920 \times 1080$ | Galaxy S20+ | TrustZone | 421.9 KB | 6,751 | 60.759s | 60.819s |
| | Galaxy S20+ | StrongBox | 451.2 KB | 7,221 | 64.989s | 65.049s |
| | Pixel 6 Pro | TrustZone | 307.2 KB | 4,916 | 44.244s | 44.304s |
| | Pixel 6 Pro | StrongBox | 310.3 KB | 4,965 | 44.685s | 44.745s |

is not publicly available. Their experiment was conducted on a $\mathsf{c4-highcpu-8}$ Google Cloud instance with four Intel Xeon PLATINUM 8581C CPUs@2.30GHz with 16 GB of RAM. Empirically, their hash verification costs about 9ms per 64-byte block, and signature verification adds about a constant 60ms. We report the end-to-end cost of verifying a single data passport in Table 1 and break down the cost model of each step as follows:

- **Hash verification:** Let $|x|$ be the encoded image size in bytes. With SHA-256 padding (one 0x80 byte and an 8-byte length), the number of 64-byte blocks is $n = \left\lceil \frac{|x|+9}{64} \right\rceil$. We report the number of blocks (**# blocks**) and hash verification time (**Hash time**) for different image resolutions, devices, and TEEs in Table 1.

- **Signature verification:** Signature verification cost is independent of the image size and adds a fixed $\approx 60$ ms prover time for each signature.

- **End-to-end passport verification:** The end-to-end passport verification cost consists of the hash and signature verification time and a lightweight consistency check. The hash verification cost scales linearly with the number of 64-byte blocks at about 9 ms per block, while signature verification adds a size-independent 60 ms. As shown in Table 1, the total prover time is dominated by hash verification, while signature verification adds only a small constant overhead. We report prover time as the verifier has negligible cost in comparison.

## 5 DISCUSSION AND FUTURE WORK

Our current prototype verifies each data passport independently, so total prover cost scales linearly with the dataset size: this is mainly due to the hash verification proof, which is inherently linear to the data size; signature verification contributes only a small constant. While this linear scaling is a limitation, we comment that the hash verification bottleneck can be avoided through parallelization: Frigo & abhi shelat (2024) reports roughly doubled prover time when generating the proof on Pixel 6 pro ($\approx 18$ ms per 64-byte block). Therefore, users could generate their own hash-consistency proofs and upload only ZK proofs plus signatures. This would significantly reduce wall-clock time without changing trust assumptions. Another future direction would be preventing the institution from selectively using user data to meet their adversarial need by enabling users to send confidential signals to the auditor.

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
