# OpenReview forum: "Data Passports: Confidentially Provable Provenance for Onboarding Verifiable ML"
_ICLR.cc/2026/Conference — Submitted to ICLR 2026_

### Official Review · Reviewer_Reco · 2025-10-27

**Soundness:** 1
**Presentation:** 1
**Contribution:** 2
**Rating:** 2
**Confidence:** 4

**Summary:**

The authors develop Data Passports - a system that aims to enable verifiable provenance by generating metadata at capture time that attests to the fact that the data was not tampered with. This attestation (denoted the "confidential passport") is then passed downstream to the institution (entity that holds the ML model), who can then employ zero-knowledge proofs to prove to an auditor that each data point is authentic (via hash consistency and signature verification).

**Strengths:**

The authors identify the issue of self-poisoning - in realistic scenarios, entities may be incentivized to modify their own data in order to circumvent requirements or policy that prevents them from making full use of their data. This is a novel and, to my knowledge, underdressed problem in the literature. To this end, the authors present a system that partially alleviates this concern.

**Weaknesses:**

System is not end-to-end: Data Passports handles the data generation phase, but training is not handled.  The paper is motivated for this purpose and without it, Data Passports serves no purpose for verifiable training.

It is unclear why TEEs are even necessary for this purpose. The signatures and ZK components of Data Passports rely on computational assumptions. I believe there may be an argument to be made for TEEs as a way to make it harder for the user (the entity collecting data) to circumvent Data Passports entirely, but users might be able to alter their behavior to avoid collecting Data Passports-backed data if they were motivated. Ideally, the paper should make it clear what purpose the TEE serves.

**Questions:**

1. The security of Data Passports relies heavily on the TEE to ensure that data was not manipulated before the passport is produced, but there is a long line of work demonstrating that TEEs are often breakable (e.g., [1]). How does this impact the security properties of your system?
2. How does the Data Passports system compare to other metadata-based provenance systems like C2PA [2]?
3. I'm not sure how this system resolves the self-poisoning issue. I understand that the system makes it hard for the institution to convince the auditor that manipulated data is from a certified data source. However, once the institution passes verification checks, what's stopping them from training models with slightly (i.e., manipulated) data in step 6? I believe this can be prevented with zero-knowledge proofs of training, but the paper itself mentions this is prohibitively expensive. This should be explained clearly as it's a major motivating point of the paper.

Presentation remarks:
1. add forward references for contributions
2. fix poor quality citations (a, b, etc)
3. Related work on ZKP is missing references
4. Row 38/39: Sentence starting with "Public verification called for" doesn't make sense
5. Row 44/45: form -> forms
6. Row 44 to 49: First half of the second intro paragraph needs to be rewritten - broken grammar, run on sentence, etc.
7. Row 98: consumptions -> consumption
8. Row 104: newborns are issued birth certificates, not necessarily passports
9. Row 122: orx -> or

[1] Van Schaik, Stephan, et al. "Sok: Sgx. fail: How stuff gets exposed." 2024 IEEE symposium on security and privacy (SP). IEEE, 2024.
[2] Rosenthol, Leonard. "C2PA: the world’s first industry standard for content provenance (Conference Presentation)." Applications of Digital Image Processing XLV. Vol. 12226. SPIE, 2022.

---

> ### Author Response · Authors · 2025-11-25
> **Thanks for your careful reading of our paper, we responded to all your questions! (Authors response 1/3)**
>
> > **there is a long line of work demonstrating that TEEs are often breakable (e.g., [1])**
>
> Indeed, there is a large body of work on attacking and hardening TEEs, but analysing these attacks are not the focus of our paper. Following your suggestion, we will add an explicit discussion of TEE vulnerabilities and justify why they are less applicable and critical in our user-side TEEs.
>
> There have been various discoveries regarding vulnerabilities of different TEEs. However, it has been shown in the literature that most of them offer limited attack capability or have high technical barriers. For example, CounterSEVeillance [Gast et al, NDSS’2026] demonstrated that a performance-counter side channel attack against TOTP was demonstrated to be possible; however, its effectiveness degrades significantly in a noisy real-world environment where the adversary loses a lot of its original capability. In another recent work [Muench et al, USENIX WOOT’2025], the physical CPU epoxy package needs to be decapsulated in order to mount the attack, which requires a decent amount of technical and hardware skills, as well as specialised equipment.
>
> In our system, TEE runs on the user device (not server-side), making it more immune to such vulnerabilities for two main reasons:
> Requirements for running such attacks. Many of the attacks against TEEs in the literature demand substantial expertise and resources which a typical user lacks.
> Incentives for running such attacks. We consider TEE assumptions on the user side and ZKP cryptographic assumptions on the server side. Users usually do not have the incentives and/or resources to bypass the security guarantees of TEE and mount attacks against their own devices, whereas the large institution is more likely to have those capabilities and incentives.
>
> As for the reason we opt to use TEE on the user side, most modern mobile devices (e.g., smartphones) are built with TEE support (for both Android and iOS). We acknowledge that TEE might not be capable of guaranteeing that all the data are uncompromised, but it will ensure that data from most users are legitimate. Institutions can further use data poisoning defences [Goldblum et al, PAMI’2022] to deal with a small fraction of corrupted data.
>
> Finally, we would like to highlight that TEE is still a best effort approach, where it does a great job on shrinking the size of Trusted Computing Base (TCB). Prior to TEE, almost all hardware and software needed to be trusted, whereas TEE only requires users to trust a small set of hardware and software primitives. It is in fact due to the strong promises given by TEE, many security researchers were drawn to find weaknesses in it. As also pointed out by the authors of the paper [1] that you cited, we expect it to get better and be more secure overtime. After all, TEE itself is not a single architecture or design, but rather a concept (of shrinking the amount of trusted components). In addition, there has been an ongoing research effort being put into developing newer and better TEEs (e.g. [Yao et al, MobiSys’2023],[Lee et al., EuroSys’2020] ).
>
>
> References:
> - [Gast et al, NDSS’2026] Stefan Gast, Hannes Weissteiner, Robin Leander Schröder, Daniel Gruss. CounterSEVeillance: Performance-Counter Attacks on AMD SEV-SNP. Network and Distributed System Security (NDSS) Symposium’ 2026.
> - [Muench et al, USENIX WOOT’2025] Marius Muench, Aedan Cullen, Kévin Courdesses, Thomas ’stacksmashing’ Roth, Andrew Zonenberg. Security through transparency: tales from the RP2350 hacking challenge. USENIX WOOT Conference on Offensive Technologies, 2025.
> - [Goldblum et al, PAMI’2022] Goldblum et al. Dataset Security for Machine Learning: Data Poisoning, Backdoor Attacks, and Defenses.IEEE Transactions on Pattern Analysis and Machine Intelligence (PAMI) 2022.
> - [Yao et al, MobiSys’2023] Zhihao Yao, Seyed Mohammadjavad Seyed Talebi†, Mingyi Chen†,Ardalan Amiri Sani†, Thomas Anderson Minimizing a Smartphone's TCB for Security-Critical Programs with Exclusively-Used, Physically-Isolated, Statically-Partitioned Hardware. International Conference on Mobile Systems, Applications and Services (MobiSys), 2023.
> - [Lee et al., EuroSys’2020] Dayeol Lee, David Kohlbrenner, Shweta Shinde, Dawn Song, Krste Asanović. Keystone: An Open Framework for Architecting TEEs. European Conference on Computer Systems (EuroSys), 2020

---

> ### Author Response · Authors · 2025-11-25
> **Authors response 2/3**
>
> > **ensure that data was not manipulated before the passport is produced**.
>
> The data is fed directly into the TEE, without giving the user or institution a chance for modification. Institutions cannot inject any code into TEE on the user device, as they are not controlled by the institution; instead, they are controlled by the manufacturer of the TEE device. Even a user itself cannot inject any code as TEEs on smartphones. For example, TrustZone and Apple Secure Enclave Processor (SEP), are designed to be fully isolated and secure from the normal execution environment, and verifiably run fixed functions and have fixed exposed APIs at the manufacturer's time, which do not allow arbitrary code to be executed. However, like any other security solution, bugs in the implementation and design flaws could result in a compromise for a capable adversary. But we note that the bar for comprising TEEs is pretty high and will continue to go higher as manufacturers mitigate the discovered vulnerabilities, especially in our design, where only a minimal manufacturer-provided “capture-and-sign” function runs inside the TEE: capturing the image and immediately signing a hash of the captured data. The data is fed directly into the TEE, without giving the user or institution a chance for modification. To modify the data after capture and before it gets fed into TEE, the attacker needs to first compromise our TCB. We attest the application and the OS using TEE primitives to ensure their authenticity and integrity.
>
> We will clarify these points in detail in the paper:
> - Where does the TEE come from? Most modern smartphones (Android and iOS) already ship with a hardware-backed TEE provided and controlled by the device manufacturer.
> - Which party installs the code inside the TEE? Device manufacturer
> - Which code runs inside TEE in our design? Only a minimal, manufacturer-provided “capture-and-sign” function: it captures the image and immediately signs a hash of the captured data.
> - How do we know this code is actually running in the TEE? Through remote attestation. The TEE produces a report (signed by the device manufacturer) that confirms the identity of the code running inside the TEE.
> - Who can verify the remote attestation report, and when? The report is verified at the consumer side (both institution and/or auditor), by checking if it is validly signed by the manufacturer or not and if all metadata contained in the report indicates a genuine device/data.
> - Which party can or cannot inject malicious code into TEE? Only the device manufacturer can inject code. Users and Institutions cannot inject arbitrary code into TEE. Even with root access in the normal world, they are still not capable of directly injecting code into TEE, which is guaranteed by hardware primitives enforced by the silicon itself.

---

> > ### Author Response · Authors · 2025-11-25
> > **Authors response 3/3**
> >
> > > **How does the Data Passports system compare to other metadata-based provenance systems like C2PA [2]?**
> >
> >
> > Thanks for your suggestion. C2PA is a technical specification, rather than a working system. Our system is actually compatible with the C2PA standard, as the cryptographic algorithms we leverage to generate a passport are included in the C2PA. Being compatible with C2PA, it enables our system to be easily integrated with other C2PA-compliant systems/platforms. Furthermore, our system incorporates ZKP for confidentiality, which C2PA does not include.
> >
> > > **what's stopping them from training models with slightly (i.e., manipulated) data in step 6? I believe this can be prevented with zero-knowledge proofs of training, but the paper itself mentions this is prohibitively expensive. This should be explained clearly as it's a major motivating point of the paper.**
> >
> > The focus of our work is *onboarding* data with provenance information into a verifiable ML pipeline. Verifiable preprocessing and training using the data can be accomplished in ZKP using previous work without invalidating passports (e.g. [1],[2]). In settings where the computational costs for end-to-end cryptographically verified training are too high, passported data could be used in conjunction with previous work on cryptographic verification of e.g. group fairness [3] and empirical calibration [4] to construct authenticated pools of data for auditing. This would address known problems in existing work that stem from reference set selection.
> >
> > The alternative is to perform all pre-processing on the device and sign the pre-processed data (instead of raw data) using TEE.
> >
> > > **Presentation remarks**
> > Thanks for your careful reading, we incorporated all of your suggestions.
> >
> >
> > We will make this clear by expanding on the related discussion at the end of Section 3.
> >
> > References:
> > [1] Xiaoyu Fan et al. PPCA: Privacy-preserving Principal Component Analysis Using Secure Multiparty Computation(MPC). 2021.
> > [2] Haochen Sun et al. zkDL: Efficient Zero-Knowledge Proofs of Deep Learning Training. 2024.
> > [3] Shehar Segal et al. Fairness in the Eyes of the Data: Certifying Machine-Learning Models. 2021.
> > [4] Stephan Rabanser et al. Confidential Guardian: Cryptographically Prohibiting the Abuse of Model Abstention. 2025.

---

### Official Review · Reviewer_yN6w · 2025-10-29

**Soundness:** 2
**Presentation:** 3
**Contribution:** 2
**Rating:** 4
**Confidence:** 5

**Summary:**

This paper tackles a key gap in trustworthy ML pipelines: pre-commitment data tampering, where training data can be manipulated before it’s cryptographically committed for verifiable training. The authors propose Data Passports, a framework combining Trusted Execution Environments (TEEs) and Zero-Knowledge Proofs (ZKPs) to ensure end-to-end verifiable data provenance. In the proposed system, data generated on user devices (e.g., smartphones) is cryptographically signed within a TEE at capture time, producing a "passport" that proves authenticity. Institutions then use ZKPs to prove to auditors that their training datasets consist solely of passported data—preserving confidentiality while ensuring provenance. The authors implement TEE-based passport creation on Android devices and estimate the institution-side ZKP verification costs using parameters from prior work. They report minimal user-side overhead and extrapolate ZKP verification times in the range of tens to hundreds of seconds per image.

**Strengths:**

+ The paper identifies a critical and timely vulnerability of pre-commitment data tampering in the verifiable ML pipeline
+ The paper effectively integrates TEEs and ZKPs to establish a verifiable, confidential chain of data provenance.
+ The user-side evaluation provides strong evidence on real Android devices that TEE-based data passport generation is efficient and feasible.

Overall, the paper is well written and easy to follow. The idea of combining ZKPs with authenticated data sources is conceptually interesting and, if further developed, could inspire future research in trustworthy and verifiable ML.

**Weaknesses:**

The paper, while well-intentioned, suffers from several critical weaknesses that undermine its core claims.

- My major concern arises from the evaluation of the core ZKP component. It appears the authors did not implement their ZKP circuit but instead chose to "extrapolate costs" from a prior paper [Frigo & abhi shelat, 2024]. Since ZKP costs depend heavily on circuit structure, memory layout, and batching strategy, this extrapolation is not a reliable proxy for end-to-end feasibility. The reported prover times (40–290s per image) imply tens of days of computation for even modest datasets (10^5 images). This may render the system infeasible for its intended ML onboarding use case in practice. A minimal working prototype, even for small circuits, would have strengthened the paper's empirical foundation.

- A second major concern lies with the security, which relies on overly optimistic trust assumptions and omits key threats related to the TEE use. The paper emphasizes TEE efficiency and isolation benefits but overlooks their accompanying risks.
  1. The framework does not consider malicious code inside the TEE. However, an institution-controlled application could (and it is possible) inject bias or alter data within the TEE before signing, yielding a valid passport for tampered input. This could directly undermine the claimed protection against "self-poisoning".
  2. The passport only attests to where and when a capture occurred, not what was captured. A user could take a TEE-certified photo of a poisoned or synthetic image displayed on a monitor (aka analog hole attack); the system would treat it as authentic and this easily bypasses the protection against "user data manipulation".
  3. The threat model implicitly assumes a "perfect" TEE. It omits adequate discussion of real-world hardware attacks like side-channels, fault-injection, or rollback attacks, which is a significant oversight for a system whose entire trust is anchored in this hardware.

- Third, it reads as though the paper’s main technical contribution lies in this TEE–ZKP “co-design,” but it is presented more like an intuitive A+B combination than a genuinely integrated system. Prior work has already explored TEE-assisted provenance (e.g., Truepic, ProvCam, Vronicle) and ZKML frameworks in depth. What's missing here is a discussion of the practical difficulties created by this very integration itself—issues that are crucial to assessing whether the proposed approach can actually work in practice.
  1. The paper frames the ZKP cost as a simple "extrapolation", but it's a direct and fatal consequence of this design. Forcing the ZKP to prove a SHA-256 hash over the entire image file (as TEE-signing requires) is exactly why the cost is tens of seconds per image. The "co-design" itself is what makes the system a bit unscalable.
  2. The current passport only binds a raw data file $Sign_{sk}(SHA256(data))$. This is fundamentally incompatible with any real-world ML pipeline. How are data labels authenticated? What about the necessary pre-processing, data cleaning, or feature extraction steps that all ML requires? Any of these steps would invalidate the passport.

- Minor:
  1. Line 045, "various form of" -> "various forms of".
  2. Line 266, "proceeds capturing" -> "proceeds to capture".
  3. Line 352, "an passport" -> "a passport".

Overall, because it avoids these hard questions, the "ZKP-CI" primitive remains largely a conceptual sketch. It lacks formal security definitions, proofs, and practical validation of scalability, which limits its value as a concrete technical contribution.

**Questions:**

1. Could the authors elaborate on the decision to extrapolate ZKP costs rather than implementing even a simplified prototype? This makes the core feasibility claim difficult to verify.
2. Relatedly, how would the performance or complexity change if the ZKP were redesigned to avoid hashing the entire image (e.g., through commitments on feature-level data or chunked proofs)? Exploring such variants could clarify whether the scalability bottleneck is fundamental or merely an artifact of the current design.
3. How does the framework defend against a certified-but-malicious app that tampers with data inside the TEE before signing? If this threat is considered out of scope, could the authors clarify why it is not a concern for the intended use cases?
4. How does the system address the "analog hole" attack for user data poisoning? If this vector is out of scope, a discussion of its implications would help clarify the system’s real-world robustness.
5. Could the authors comment on why real-world TEE vulnerabilities (e.g., side-channels, fault-injection) were omitted from the threat model, given they directly impact the system's root of trust?
6. Since the passport currently signs only raw data, how does the system authenticate downstream processing steps—such as labeling, normalization, or feature extraction—that are essential for ML training? Without this, how can the provenance guarantee extend to the actual training dataset?

**Details Of Ethics Concerns:**

This is not a major issue, but I will lay out below anyway:

- The system assumes TEEs directly access and sign user data at capture time. Depending on deployment, this could introduce privacy risks if the data or associated metadata (e.g., geolocation, timestamps) are leaked or linked to user identity.

---

> ### Author Response · Authors · 2025-11-25
> **Thank you so much for the thoughtful review and for raising clear questions! We responded to all your questions. (Authors response 1/4)**
>
> > **Could the authors elaborate on the decision to extrapolate ZKP costs rather than implementing even a simplified prototype? This makes the core feasibility claim difficult to verify.**
>
> **ZKP circuit**.
> Our ZK relation for passport verification is exactly the standard ECDSA verification: “I know a valid ECDSA signature on the hash of this data under a certified public key.” This is the same relation implemented and optimized in the circuit provided by Frigo & shelat [2024]. We wish to clarify that we do not propose a different ZK protocol, and our design is meant to reuse their optimized circuit as a drop-in component. Therefore, our estimates are not loose extrapolations, but a direct application of SOTA benchmarks to the exact same algebraic task.
>
> Because their code is not open-sourced, faithfully re-implementing their highly optimized ECDSA circuit and prover (which includes low-level tuning) would largely be a cryptographic engineering effort, not a conceptual contribution of this work. We therefore take their published prover times and circuit sizes as a black-box building block and scale them to our setting. We will revise the paper to state more clearly that: (i) we target the same ECDSA-verification relation; (ii) our extrapolation assumes using their optimized circuit rather than a new one.
>
> **Scalability for large databases**.
> The concern regarding "tens of days" of computation assumes sequential execution. In our framework, ZKP is performed offline by the Institution, not on user devices. Because each passport is verified independently, the workload is embarrassingly parallel. A modest server cluster (e.g., 100 nodes) reduces the wall-clock time for 10^5 images from days to hours. Crucially, we did fully implement and benchmark the resource-constrained component (TEE signing on smartphones); the heavy ZK computation is deliberately offloaded to the institution where computation resources are less limited.
>
>
> > **Relatedly, how would the performance or complexity change if the ZKP were redesigned to avoid hashing the entire image (e.g., through commitments on feature-level data or chunked proofs)? Exploring such variants could clarify whether the scalability bottleneck is fundamental or merely an artifact of the current design**.
>
> Without a ZKP-verified commitment opening (fulfilled by ZKP hashing in our method), there is to our knowledge no existing way to confirm to the auditor that the institution has not modified the data obtained from the sources. However, we concur that exploring optimized variants on the original protocol is well-motivated. We outline an optimized procedure which could replace hashing below:
>
> **Optimized Passport Setup**:
> Source randomly partitions the bits of their input string $w$ into sets A and B, and computes the parity of each set $(p_A, p_B)$. Source sends an encryption of $(p_A, p_B)$, along with the indices included in A and B, to the auditor.
> Source signs $(p_A, p_B)$ in TEE, sends $w$, $(p_A, p_B)$, and $k$ to the institution.
>
> **ZKP Validation**:
> Institution commits to $w’$.
> Auditor sends index sets $A, B$ to Institution. Institution computes $(p’_A, p’_B)$, the parity of $w’$ partitioned over $A$ and $B$. It proves in zero-knowledge that $(p’_A, p’_B) = (p_A, p_B)$. If the proof fails, auditor aborts.
>
> If the Institution committed to $w’ \neq w$, we can show that the probability that $(p’_A, p’_B) = (p_A, p_B)$ passes is at most 1/2, since including any odd number of altered bits from $w$ in $A$ or $B$ will alter the parity. Thus performing this protocol over $\lambda$ partitions results in a $2^{-\lambda}$ success rate for a cheating adversary. We note also that parity is much cheaper to verify in ZKP than SHA-256, thus this will substantially optimize the protocol.

---

> ### Author Response · Authors · 2025-11-25
> **Authors response 2/4**
>
> > **How does the framework defend against a certified-but-malicious app that tampers with data inside the TEE before signing? If this threat is considered out of scope, could the authors clarify why it is not a concern for the intended use cases?**
>
> Institutions cannot inject any code into TEE on the user device, as they are not controlled by the institution; instead, they are controlled by the manufacturer of the TEE device. Even a user itself cannot inject any code as TEEs on smartphones. For example, TrustZone and Apple Secure Enclave Processor (SEP), are designed to be fully isolated and secure from the normal execution environment, and verifiably run fixed functions and have fixed exposed APIs at the manufacturer's time, which do not allow arbitrary code to be executed. However, like any other security solution, bugs in the implementation and design flaws could result in a compromise for a capable adversary. But we note that the bar for comprising TEEs is pretty high and will continue to go higher as manufacturers mitigate the discovered vulnerabilities, especially in our design, where only a minimal manufacturer-provided “capture-and-sign” function runs inside the TEE: capturing the image and immediately signing a hash of the captured data. The data is fed directly into the TEE, without giving the user or institution a chance for modification. To modify the data after capture and before it gets fed into TEE, the attacker needs to first compromise our TCB. We attest the application and the OS using TEE primitives to ensure their authenticity and integrity.
>
> We will clarify these points in detail in the paper:
> - Where does the TEE come from? Most modern smartphones (Android and iOS) already ship with a hardware-backed TEE provided and controlled by the device manufacturer.
> - Which party installs the code inside the TEE? Device manufacturer
> - Which code runs inside TEE in our design? Only a minimal, manufacturer-provided “capture-and-sign” function: it captures the image and immediately signs a hash of the captured data.
> - How do we know this code is actually running in the TEE? Through remote attestation. The TEE produces a report (signed by the device manufacturer) that confirms the identity of the code running inside the TEE.
> - Who can verify the remote attestation report, and when? The report is verified at the consumer side (both institution and/or auditor), by checking if it is validly signed by the manufacturer or not and if all metadata contained in the report indicates a genuine device/data.
> - Which party can or cannot inject malicious code into TEE? Only the device manufacturer can inject code. Users and Institutions cannot inject arbitrary code into TEE. Even with root access in the normal world, they are still not capable of directly injecting code into TEE, which is guaranteed by hardware primitives enforced by the silicon itself.
>
>
> > **How does the system address the "analog hole" attack for user data poisoning? If this vector is out of scope, a discussion of its implications would help clarify the system’s real-world robustness.**
>
> Thanks for the good suggestion. Liu et.al. [Sccop, USENIX2025], systematically studied the recapture (analog hole) attack that you thoughtfully suggested and proposed a systematic mitigation strategy by leveraging state-of-the-art depth sensing technologies as well as learning-based depth estimation.
>
> We will describe the attack and analyse the defence as being complementary building block to ours in Section 3:
> Attack (recapture/analog hole). Liu et.al. [Sccop, USENIX2025] studied a recapture attack where the adversary displays fake content on some form of a screen (e.g., TV, projector, or computer screen) or surface (e.g., cardboard, canvas, or paper) and uses a provenance-asserting secure camera device to capture photos and videos of the displayed content. The resulting media carries valid provenance but does not reflect an authentic real-world scene.
> Defence (Scoop). Liu et.al. [Sccop, USENIX2025] proposed Scoop to detect misleading recaptures, i.e., a recaptured photo or video where the presence of a display medium is not visually identifiable. Scoop fuses Time-of-Flight (ToF) depth sensor data with ML-based monocular depth estimation to detect physical and logical inconsistencies indicative of a screen or recapture.
>
> [Sccop, USENIX2025] Yuxin Liu, Habiba Farrukh, and Ardalan Amiri Sani, UC Irvine; Sharad Agarwal, Microsoft; Gene Tsudik, UC Irvine. Scoop: Mitigation of Recapture Attacks on Provenance-Based Media Authentication, USENIX2025

---

> > ### Author Response · Authors · 2025-11-25
> > **Authors response 3/4**
> >
> > > **Could the authors comment on why real-world TEE vulnerabilities (e.g., side-channels, fault-injection) were omitted from the threat model, given they directly impact the system's root of trust?**
> >
> > Indeed, there is a large body of work on attacking and hardening TEEs, but analysing these attacks are not the focus of our paper. Following your suggestion, we will add an explicit discussion of TEE vulnerabilities and justify why they are less applicable and critical in our user-side TEEs.
> >
> > There have been various discoveries regarding vulnerabilities of different TEEs. However, it has been shown in the literature that most of them offer limited attack capability or have high technical barriers. For example, CounterSEVeillance [Gast et al, NDSS’2026] demonstrated that a performance-counter side channel attack against TOTP was demonstrated to be possible; however, its effectiveness degrades significantly in a noisy real-world environment where the adversary loses a lot of its original capability. In another recent work [Muench et al, USENIX WOOT’2025], the physical CPU epoxy package needs to be decapsulated in order to mount the attack, which requires a decent amount of technical and hardware skills, as well as specialised equipment.
> >
> > In our system, TEE runs on the user device (not server-side), making it more immune to such vulnerabilities for two main reasons:
> > Requirements for running such attacks. Many of the attacks against TEEs in the literature demand substantial expertise and resources which a typical user lacks.
> > Incentives for running such attacks. We consider TEE assumptions on the user side and ZKP cryptographic assumptions on the server side. Users usually do not have the incentives and/or resources to bypass the security guarantees of TEE and mount attacks against their own devices, whereas the large institution is more likely to have those capabilities and incentives.
> >
> > As for the reason we opt to use TEE on the user side, most modern mobile devices (e.g., smartphones) are built with TEE support (for both Android and iOS). We acknowledge that TEE might not be capable of guaranteeing that all the data are uncompromised, but it will ensure that data from most users are legitimate. Institutions can further use data poisoning defences [Goldblum et al, PAMI’2022] to deal with a small fraction of corrupted data.
> >
> > Finally, we would like to highlight that TEE is still a best effort approach, where it does a great job on shrinking the size of Trusted Computing Base (TCB). Prior to TEE, almost all hardware and software needed to be trusted, whereas TEE only requires users to trust a small set of hardware and software primitives. It is in fact due to the strong promises given by TEE, many security researchers were drawn to find weaknesses in it. After all, TEE itself is not a single architecture or design, but rather a concept (of shrinking the amount of trusted components). In addition, there has been an ongoing research effort being put into developing newer and better TEEs (e.g. [Yao et al, MobiSys’2023],[Lee et al., EuroSys’2020] ).
> >
> > References:
> > - [Gast et al, NDSS’2026] Stefan Gast, Hannes Weissteiner, Robin Leander Schröder, Daniel Gruss. CounterSEVeillance: Performance-Counter Attacks on AMD SEV-SNP. Network and Distributed System Security (NDSS) Symposium’ 2026.
> > - [Muench et al, USENIX WOOT’2025] Marius Muench, Aedan Cullen, Kévin Courdesses, Thomas ’stacksmashing’ Roth, Andrew Zonenberg. Security through transparency: tales from the RP2350 hacking challenge. USENIX WOOT Conference on Offensive Technologies, 2025.
> > - [Goldblum et al, PAMI’2022] Goldblum et al. Dataset Security for Machine Learning: Data Poisoning, Backdoor Attacks, and Defenses.IEEE Transactions on Pattern Analysis and Machine Intelligence (PAMI) 2022.
> > - [Yao et al, MobiSys’2023] Zhihao Yao, Seyed Mohammadjavad Seyed Talebi†, Mingyi Chen†,Ardalan Amiri Sani†, Thomas Anderson Minimizing a Smartphone's TCB for Security-Critical Programs with Exclusively-Used, Physically-Isolated, Statically-Partitioned Hardware. International Conference on Mobile Systems, Applications and Services (MobiSys), 2023.
> > - [Lee et al., EuroSys’2020] Dayeol Lee, David Kohlbrenner, Shweta Shinde, Dawn Song, Krste Asanović. Keystone: An Open Framework for Architecting TEEs. European Conference on Computer Systems (EuroSys), 2020

---

> > > ### Author Response · Authors · 2025-11-25
> > > **Authors response 4/4**
> > >
> > > > **Since the passport currently signs only raw data, how does the system authenticate downstream processing steps—such as labeling, normalization, or feature extraction—that are essential for ML training? Without this, how can the provenance guarantee extend to the actual training dataset?**
> > >
> > > The focus of our work is *onboarding* data with provenance information into a verifiable ML pipeline. Verifiable preprocessing and training using the data can be accomplished in ZKP using previous work without invalidating passports (e.g. [1],[2]). In settings where the computational costs for end-to-end cryptographically verified training are too high, passported data could be used in conjunction with previous work on cryptographic verification of e.g. group fairness [3] and empirical calibration [4] to construct authenticated pools of data for auditing. This would address known problems in existing work that stem from reference set selection.
> > >
> > > The alternative is to perform all pre-processing on the device and sign the pre-processed data (instead of raw data) using TEE.
> > >
> > > We will make this clear by expanding on the related discussion at the end of Section 3.
> > >
> > > References:
> > > - [1] Xiaoyu Fan et al. PPCA: Privacy-preserving Principal Component Analysis Using Secure Multiparty Computation(MPC). 2021.
> > > - [2] Haochen Sun et al. zkDL: Efficient Zero-Knowledge Proofs of Deep Learning Training. 2024.
> > > - [3] Shehar Segal et al. Fairness in the Eyes of the Data: Certifying Machine-Learning Models. 2021.
> > > - [4] Stephan Rabanser et al. Confidential Guardian: Cryptographically Prohibiting the Abuse of Model Abstention. 2025.

---

> > > > ### Comment · Reviewer_yN6w · 2025-11-28
> > > >
> > > > Thank you for your detailed response and the effort put into the rebuttal. I have read your clarifications; But in my opinion, several core concerns regarding the practical feasibility and system design remain unaddressed. Therefore, I will maintain my original score.
> > > >
> > > > - The argument that "parallelization" solves the scalability issue addresses latency but ignores the prohibitive total computational cost (validating a mere $10^5$ images requires about 50 days of single-CPU time). Furthermore, the introduction of the "Parity check" protocol in the rebuttal implicitly admits that the original design (SHA256 in ZKP) was indeed a bottleneck. To me, this appears to lead to a completely new (and not-yet-ready-to-be-verified) design. Since it is not included in the original submission, I am not convinced that the argument could serve as a basis for acceptance.
> > > >
> > > > - The proposed defense against "analog hole" attacks relies on external works (e.g., Scoop, USENIX '25) and specific hardware (ToF), which effectively shifts the solution to a largely new, unreviewed system design. Similarly, the argument that TEE code is "fixed" ignores the broader attack surface: the software stack interacting with the TEE remains a vector for potential self poisoning and manipulation, which the current threat model fails to address adequately.
> > > >
> > > > - Confirming that the TEE only signs raw data exposes a fatal semantic gap: neither labels nor essential preprocessing steps (e.g., resizing, normalization) are cryptographically bound to the data. This means the "provenance" is technically lost the moment raw data is transformed for actual training. The authors' suggestion to push these heavy preprocessing and labeling verification steps to the ZKP layer, which is already the system's primary performance bottleneck, is an architectural contradiction that further degrades feasibility. Thus, I consider the argument not strong enough to change my mind.

---

### Official Review · Reviewer_QHC2 · 2025-10-30

**Soundness:** 2
**Presentation:** 2
**Contribution:** 2
**Rating:** 0
**Confidence:** 4

**Summary:**

The paper proposes “Data Passports” as a mechanism to certify training-data provenance for machine‐learning pipelines. Each user‐generated data item is issued a cryptographic “passport” at creation time (within a Trusted Execution Environment, TEE) encoding provenance metadata. The institution then uses a zero-knowledge proof (ZKP) to show an auditor that it only used data items bearing valid passports — thus preventing both institutional data manipulation (post-commit tampering) and user-side poisoning of datasets. The authors claim this extends verifiable ML beyond just model training to *data authenticity*, and provide a prototype implementation (on Android phones + ZKP verification) to show low overhead.

**Strengths:**

Tackles a timely and relevant problem: data provenance is increasingly recognised as a weak link in trustworthy ML.

The authors include a prototype and some empirical overhead measurements (on Android devices, CPU/memory/battery of passport creation) which strengthens the work’s engineering credibility.

The intuitive analogy of a “passport for data” helps make the idea accessible, which is helpful given the intersection of ML, security, and hardware trust.

**Weaknesses:**

Justification of ZKP & TEE is weak. The paper does not convincingly explain *why* a full zero‐knowledge proof is required, or *why* a TEE is needed rather than a simpler approach (e.g., device signs the hash of the sample, institution publishes signature, verifier checks signature). If the passport metadata are non‐sensitive, then a simpler digital signature scheme might suffice. The use of ZKP and TEE therefore feels like over‐engineering or lacking sufficient motivation/discussion.

Bias and provenance vs. authenticity trade‐off omitted. While the mechanism aims at authenticity, it doesn’t address *bias*, representativeness or *quality* of the data. The system might certify “this sample was captured by a trusted device at time T” but that says nothing about whether the sample is training‐appropriate, non-biased, or non‐poisonous (in a semantic sense). The authors should more clearly delineate what the passport does *and does not* guarantee.

Security model is insufficiently detailed. The paper lacks a formal adversary model, threat‐analysis table, or proof sketch of soundness. It assumes the TEE, certificate authority (CA), and device registration are trusted, but does not articulate what happens if keys are compromised, certificates forged, TEE is subverted, or side‐channels exploited. Without this, the “tamper‐proof” claim is overstated.

Clarity and writing issues. Some language is informal to me (e.g., on p.4: “I uses a zero‐knowledge proof to verify to V that …”), which detracts from readability and professionalism. Some design assumptions and implementation details are glossed too lightly.

Scalability and deployment issues under-explored. While user‐device overhead is measured, the institution-side proof generation and verification overhead for large datasets appears to grow linearly (each sample’s metadata must be proven/verified). The paper acknowledges this but does not give quantitative limits (e.g., how many millions of samples is feasible). Also, the ecosystem dependence (every device must support TEE signing; every source must be registered; legacy data without passports must be handled) is only briefly discussed.

**Questions:**

My questions are mainly listed in the part of weaknesses.

---

### Author Response · Authors · 2025-11-25
**Summary for all reviewers, thank you!**

Dear all reviewers,

We thank you for your careful reading and noting the strengths of our paper, namely:
- A timely, novel and unaddressed problem
- The intuitive analogy of a “passport for data”
- Identifying a critical and timely vulnerability of pre-commitment data tampering / the issue of self-poisoning in realistic scenario
- A prototype on real Android devices / Realistic scenario
- Interesting combination of ZKP with authenticated data sources
- The potential to inspire future research in trustworthy and verifiable ML

We respond below to all questions raised and summarise the changes here:
- Discussed why we reused the optimised circuit provided by Frigo & shelat [2024] for the standard ECDSA verification as a drop-in component.
- Discussed in detail the big literature on attacking and hardening TEEs
- Clarified that neither institution nor users can inject any malicious code into the user-device TEE controlled by the device manufacturer
- Presented an already published paper on defending against analog hole attack
- Clarified that the focus of our work is *onboarding* data with provenance information into a verifiable ML pipeline, and the fact that verifiable preprocessing and training using the data can be accomplished in ZKP using previous work

If you have any further questions, comments or concerns, we would be glad to respond promptly.

Thank you,

Authors

---

### Meta-Review · Area_Chair_FLsU · 2026-01-07

**Summary:**

The paper proposes "data passports" to certify training-data provenance in ML systems. The topic is timely in the context of trustworthy AI, the solution leverages solutions from cryptography such as zero knowledge proofs (ZKP).

While the paper tackles an important issues, the execution is not matching the expectations as highlighted by the reviewers. Except one negligent reviewer (of which the review was ignored by the authors, and in my decision), the engaged reviewers were not convinced by the merits of the paper and provided valid reasons for rejection., in particular, on the practicality and the system design part of the paper. I hope the authors carefully take into account these reviews in their future versions.

**Reviewer Concerns:**

reviewer yN6w's core concerns on practicality and system design remain unresolved

**Reviewer Scores:**

QHC2 should have (I hope) retracted some statement in their review, and offered a human-generated review…

---

### Decision · Program_Chairs · 2026-01-26

Reject